Journal of Machine Learning Research 1 (2000) 1-48                    Submitted 4/00; Published 10/00

# Weakly Supervised Global-Local Feature Learning for Cervical Cytology Image Analysis

**Editor:** Leslie Pack Kaelbling

## Abstract

Existing supervised Convolutional Neural Network (CNN) approaches for cervical cytology image analysis generally rely on the heavy manual annotation for each cell or cell mass and thus lead to extensive time and effort. In this paper, we propose a weakly supervised global-local learning for cervical cytology image analysis. It aims to perform the classification for region of interests (ROIs) and further classify the cells only with the ROI labels. Specifically, the proposed method firstly detects the cells within ROI and extracts the CNN features of cells. Then attention-based bidirectional LSTM (Att-BLSTM) is applied to explore the global contextual information of ROI. On the other hand, the Vision Transformer (ViT) is used to exploit the local attentive representations of the cells in ROIs. The cross attention (CA) is applied to incorporate the global contextual features and local patterns and thus generates more discriminative feature representation of ROI. More importantly, the CA score is used as the pseudo label to select top and least attentive cells. Therefore, the in-the-class and out-of-the-class CA branches are trained to achieve the cell classification. Experimental results demonstrate the effectiveness of our method for cervical cytology ROI and cell classification, and the weak supervision of the image-level label has great potential to promote the automatic whole slide cervical image analysis and alleviate the workload of cytologists.

**Keywords:**   Cervical cancer screening, cervical cytology, weakly supervised learning, cervical cell classification

## 1. Introduction

As the most common screening of cervical cancer at early stage, cervical cytology screening requires cytologists to observe the morphological change of cells in the Papanicolaou (PAP) smears under the microscope, then identify the normal and abnormal cell types and make the final diagnosis on the basis of The Bethesda System (TBS). However, the screening process is usually time-consuming and labor-intensive. Therefore, the automatic screening methods have been proposed to intelligently identify the abnormal cells in the cytology slide and thus reduce the workload of cytologists.

The traditional screening methods generally use the hand-crafted features and classifier to determine the cell types. However, the performance of these methods is easily limited by the feature or classifier selection. With the development of artificial intelligence, Convolutional Neural Networks (CNNs) achieve the promising performance in object detection, semantic segmentation and image classification. Gradually CNN has been applied in cervical cytology screening. Zhang et al. [Zhang et al. (2017)] use CNN to extracts deep features of cell image patch for cervical cell classification. Plissiti et al. [Plissiti et al. (2018)] intro-

duce the public cervical cell image dataset SIPaKMeD and use VGG to achieve the better classification performance compared with the handcrafted features. Shi et al. [Shi et al. (2021)] propose the cervical cell image classification method based on graph convolutional network (GCN) to exploit the potential relationship of cervical cell images. Furthermore, some studies on the whole slide image (WSI)-based cervical cell screening have attracted much attention. Chen et al. [Chen et al. (2021)] model the cervical cell as the unit and use the unit stochastic selection and attention fusion to achieve the WSI-based diagnosis. However, all these methods rely on the heavy manual cell-level annotation. Therefore, it is likely to cost extensive time and effort and easily generate the noisy samples due to the subjective differences of cytologists.

In this paper, we propose a novel cervical cytology image analysis method with weakly supervised global-local feature learning. Unlike the above methods, we achieve the region of interests (ROIs) and cell classification in a weakly supervised way. The ROI label is only used in the feature learning and thus alleviate the workload of manual annotation. Concretely the cells are firstly detected and the corresponding CNN features can be gained. Then the cell features are used as the sequential data and fed into the attention-based bi-directional LSTM (Att-BLSTM) [Zhou et al. (2016)] for capturing the global contextual information of ROI. Meanwhile, the Vision Transformer (ViT) [Dosovitskiy et al. (2020)] is performed on the feature maps of cells for exploring the local attentive representations. The cross attention (CA) is used to incorporate the global contextual features and local attentive features. Therefore, the more discriminative feature representation of ROI can be obtained for ROI classification. More importantly, CA score is taken as the pseudo label to select top and least attentive cells. Benefit from the weakly supervised setting of clustering-constrained-attention multiple-instance learning (CLAM) [Lu et al. (2021)], we train the in-the-class and out-of-the-class CA branches with the selected cells for the cell classification. Experimental results demonstrate the feasibility and effectiveness of our method for cervical cytology ROI and cell classification.

The contribution and novelty of this paper can be summarized as follows:

1. We introduce a weakly supervised global-local learning for cervical cytology image analysis. To the best of our knowledge, this is the first to apply LSTM and Transformer architecture for cervical ROI-level classification. More importantly, a weakly supervised learning strategy based on cross attention is applied to achieve the cell-level classification. The classification performance is promising compared with supervised CNN methods. It is beneficial to alleviate the heavy manual cell-level annotation.

2. We take the CNN features of cells as the sequential data and use the attention-based bi-directional LSTM (Att-BLSTM) to capture global contextual features in the ROIs. Meanwhile, the Vision Transformer (ViT) is applied to explore the local attentive features of cells. The global-local features are incorporated to enhance the representational power of ROI features.

3. We conduct the experiments on the Motic liquid-based cytology ROI dataset including 3 categories: Negative for Intraepithelial Lesion for Malignancy (NILM), low-grade squamous intraepithelial lesion (LSIL) and high-grade squamous intraepithelial lesion (HSIL). Experimental results demonstrate the effectiveness of our method and also provide the opportunities for WSI-based screening.

## 2. Methodology

### 2.1 Overview

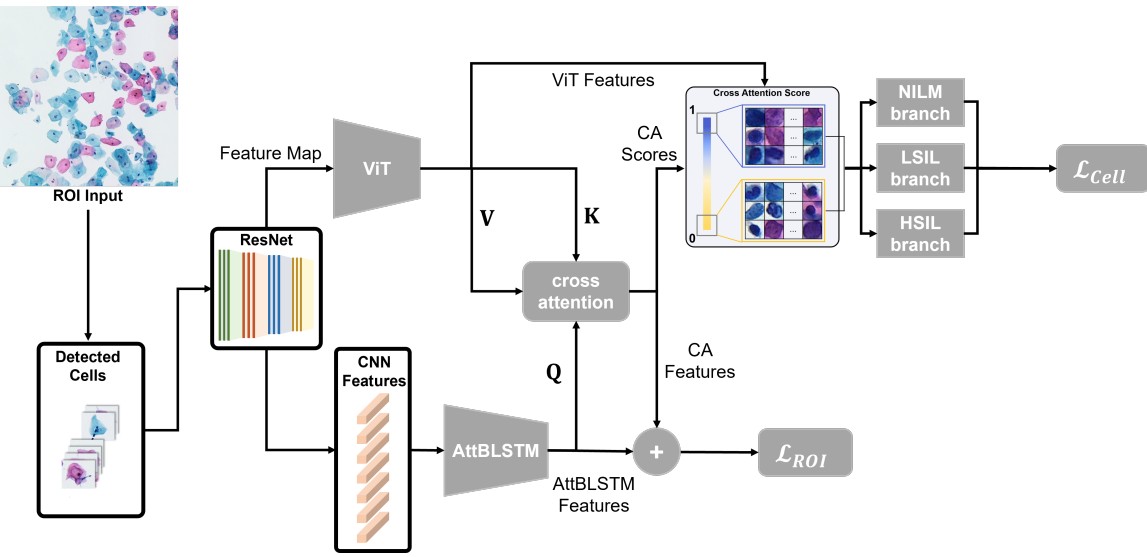

Figure 1: Pipeline of our method for cervical cell image analysis.

The pipeline of the proposed method is illustrated in Fig. 1. The cervical cells are firstly detected by YOLOv4 [Bochkovskiy et al. (2020)] and then the corresponding CNN features of cells extracted by ResNet-50 [He et al. (2016)] pre-trained on the ImageNet are fed into Att-BLSTM [Zhou et al. (2016)] for capturing the global contextual information of ROI. On the other hand, the feature maps of Stage 4 from ResNet-50 are inputted to the hybrid architecture of ViT [Dosovitskiy et al. (2020)] pre-trained on the ImageNet-21k and thus the local attentive features of cells can be gained. Afterwards, the cross attention (CA) is applied to highlight the representative cells which builds the connections between the ViT features of cells and the AttBLSTM features of ROI. Specifically the ViT features after linear transformation are respectively the key $\mathbf{K}$ and value $\mathbf{V}$ inputs of CA, and the AttBLSTM features after linear transformation taken as the query $\mathbf{Q}$. Consequently, the final ROI feature representation is generated by the sum operation of CA features and AttBLSTM features. Meanwhile, the $N$ top and $N$ least attentive cells are selected by the sorted CA scores, and respectively regarded as the positive and negative samples. Similar to CLAM [Lu et al. (2021)], given the number of classes $C$, we construct $C$ branches and each branch has a binary classification layer which predicts whether the sample belongs to this class and the corresponding probability. Note that the in-the-class CA branch corresponds the true label $y$ of given ROI and the remaining are the out-of-the class branches. For the in-the-class branch, the ViT features of $N$ positive and $N$ negative samples are all fed. Particularly, the ViT features of $N$ positive samples are taken as the negative samples of the out-of-the class branch. The entire network is updated by a hybrid loss consisting of the ROI classification loss $L_{ROI}$ and the cell classification loss $L_{cell}$ which has $C$ class-specific cross-entropy loss functions.

## 2.2 Global Att-BLSTM Feature

Inspired by the success of RNN modelling on the image patches of WSI [Zheng et al. (2020)], we firstly use pre-trained ResNet-50 to extract the features of detected cells and then model the cell features as a sequential data. Assuming $n_c$ detected cells are represented by $\mathbf{X} = [\mathbf{x}_1, \mathbf{x}_2, ..., \mathbf{n}_c] \in R^{d_f \times n_c}$ where $d_f$ is the feature dimension of ResNet-50. Att-BLSTM [Zhou et al. (2016)] is performed on the sequential cell features $\mathbf{X}$ to obtain the ROI representation. The basic structure of LSTM is formulated as [Zhou et al. (2016)]:

$$
\begin{aligned}
\mathbf{i}_t &= \sigma(\mathbf{W}_{xi}\mathbf{x}_t + \mathbf{W}_{hi}\mathbf{h}_{t-1} + \mathbf{W}_{ci}\mathbf{c}_{t-1} + \mathbf{b}_i) \\
\mathbf{f}_t &= \sigma(\mathbf{W}_{xf}\mathbf{x}_t + \mathbf{W}_{hf}\mathbf{h}_{t-1} + \mathbf{W}_{cf}\mathbf{c}_{t-1} + \mathbf{b}_f) \\
\mathbf{g}_t &= tanh((\mathbf{W}_{xc}\mathbf{x}_t + \mathbf{W}_{hc}\mathbf{h}_{t-1} + \mathbf{W}_{cc}\mathbf{c}_{t-1} + \mathbf{b}_c)) \\
\mathbf{c}_t &= \mathbf{i}_t\mathbf{g}_t + \mathbf{f}_t\mathbf{c}_{t-1} \\
\mathbf{o}_t &= \sigma(\mathbf{W}_{xo}\mathbf{x}_t + \mathbf{W}_{ho}\mathbf{h}_{t-1} + \mathbf{W}_{co}\mathbf{c}_t + \mathbf{b}_o) \\
\mathbf{h}_t &= \mathbf{o}_t tanh(\mathbf{c}_t)
\end{aligned}
\tag{1}
$$

where $\mathbf{i}_t$ is the input gate with the corresponding weight matrix $\mathbf{W}_{xi}$, $\mathbf{W}_{hi}$, $\mathbf{W}_{ci}$ and $\mathbf{b}_i$, $\mathbf{f}_t$ is the forget gate with the corresponding weight matrix $\mathbf{W}_{xf}$, $\mathbf{W}_{hf}$, $\mathbf{W}_{cf}$ and $\mathbf{b}_f$, $\mathbf{o}_t$ is the output gate with the corresponding weight matrix $\mathbf{W}_{xo}$, $\mathbf{W}_{ho}$, $\mathbf{W}_{co}$ and $\mathbf{b}_o$, $\mathbf{c}_t$ is the current cell state, $\mathbf{h}_t$ is the hidden state of the current time step $t(\mathbf{h}_0 = \mathbf{0})$ and $\sigma$ denotes the sigmoid activation function.

Considering the bidirectional LSTM (BLSTM) has two sub-networks for two directions, the output of BLSTM is represented by:

$$
\mathbf{h}_t = \lfloor \overrightarrow{\mathbf{h}_t} \bigoplus \overleftarrow{\mathbf{h}_t} \rfloor
\tag{2}
$$

$\overrightarrow{\mathbf{h}_t}$ and $\overleftarrow{\mathbf{h}_t}$ denote the output of the forward LSTM and backward LSTM respectively. To focus on the most important semantic information in the sequential data, the attention mechanism is used in Att-BLSTM and the final feature representation of ROI $\mathbf{h}_{ROI}$ can be gained by Eq. (3):

$$
\begin{aligned}
\alpha &= softmax(\mathbf{w}^T tanh(\mathbf{H})) \\
\mathbf{h}_{ROI} &= tanh(\mathbf{H}\alpha^T)
\end{aligned}
\tag{3}
$$

where $\mathbf{H}$ is the output matrix of BLSTM and $\mathbf{w}$ is the trained parameter vector. Consequently, the global contextual information of ROI can be explored by the Att-BLSTM.

## 2.3 Local ViT Feature

To exploit the local attentive representations of the cells in ROIs for improving the ROI feature representation ability and further achieve the cell classification in a weakly supervised way, ViT [Dosovitskiy et al. (2020)] is performed on the detected cells. Instead of using the raw image patches, the feature maps of Stage 4 from ResNet-50 are fed into the hybrid architecture of ViT pre-trained on the ImageNet-21k. Namely, the patches can be regarded as have the spatial size $1 \times 1$ [Dosovitskiy et al. (2020)].

The ViT has $L$ Transformer encoders which include multiheaded self-attention (MSA), Layernorm (LN) and MLP blocks, which can be characterized as follows:

$$
\begin{aligned}
\mathbf{z}_0 &= [\mathbf{b}_{class}; \mathbf{b}_1\mathbf{E}; \mathbf{b}_2\mathbf{E}, ..., \mathbf{b}_m\mathbf{E}] + \mathbf{E}_{pos}, \mathbf{E} \in R^{d_c \times D}, \mathbf{E}_{pos} \in R^{(m+1) \times D} \\
\mathbf{z}_l' &= MSA(LN(\mathbf{z}_{l-1})) + \mathbf{z}_{l-1}, l = 1, ..., L \\
\mathbf{z}_l &= MLP(LN(\mathbf{z}_l')) + \mathbf{z}_l', l = 1, ..., L \\
\mathbf{y} &= LN(\mathbf{z}_L^0)
\end{aligned}
\tag{4}
$$

where $\mathbf{b}_i (i = 1, ..., m)$ denotes the 1×1 patches, $\mathbf{z}_0^0 = \mathbf{b}_{class}$ is a learnable embedding to the input sequence, $d_c$ is the channels of the feature maps, and $\mathbf{E}$ is the learnable embedding matrix which project the patches into the $D$ dimensional latent representation of the Transformer, $\mathbf{E}_{pos}$ is the position embeddings, MLP has two layers with GELU activation function and $\mathbf{y}$ is the final representation of cell image.

## 2.4 Global-Local Feature Learning

After the ROI feature $\mathbf{h}_{ROI} \in R^{1024}$ extracted by the Att-BLSTM and the ViT features of cells $\mathbf{Y} = [\mathbf{y}_1, \mathbf{y}_2, ..., \mathbf{y}_{n_c}] \in R^{n_c \times D}$ ($D$=768) are obtained, the cross attention (CA) is used to build the connections between the ViT features of cells and the AttBLSTM features of ROI, which can highlight the representative cells and thus improve the discriminant ability of ROI feature. Note that $n_c$ denotes the number of detected cells. The cross attention can be formulated:

$$
\begin{aligned}
\mathbf{a} = Attention(\mathbf{Q}, \mathbf{K}, \mathbf{V}) &= Attention(\mathbf{h}_{ROI}\mathbf{W}^Q, \mathbf{Y}\mathbf{W}^K, \mathbf{Y}\mathbf{W}^V) \\
&= softmax(\frac{(\mathbf{h}_{ROI}\mathbf{W}^Q)(\mathbf{Y}\mathbf{W}^K)^T}{\sqrt{d_k}})(\mathbf{Y}\mathbf{W}^V)
\end{aligned}
\tag{5}
$$

where $\mathbf{W}^Q \in R^{d_f \times d_k}$, $\mathbf{W}^K \in R^{D \times d_k}$, $\mathbf{W}^V \in R^{D \times d_v}$, and $d_f = d_k = d_v = 1024$. The output $\mathbf{a}$ is regarded as the weighted sum of the cell features. The final ROI feature $\tilde{\mathbf{h}}$ can be gained by encoding the local cell features into the global contextual features, as shown in Eq.(6), and it can be used to train the cross-entropy loss $L_{ROI}$ for classifying the ROIs.

$$
\tilde{\mathbf{h}} = \mathbf{h}_{ROI} + \mathbf{a}
\tag{6}
$$

Furthermore, the CA score is defined as follows:

$$
\mathbf{s} = softmax\Big(\frac{(\mathbf{h}_{ROI}\mathbf{W}^Q)(\mathbf{Y}\mathbf{W}^K)^T}{\sqrt{d_k}}\Big)
\tag{7}
$$

which can characterize the contribution of cells to the global ROI. Therefore, motivated by the weakly supervised setting of CLAM [Lu et al. (2021)], the CA score is taken as the pseudo label to weakly supervise the cell classification. Concretely, $N$ top and $N$ least attentive cells are selected to be the positive and negative samples according to the sorted CA score. Then we build $C$ branches and each branch has a binary classification layer which predicts whether the sample belongs to this class and the corresponding probability.

Moreover, the class-specific cross-entropy loss function corresponding to each branch is used to train the binary classification layer. Note that we only consider 3 categories ($C = 3$) of cervical cell in this paper, namely NILM, LSIL and HSIL. Similar to CLAM, we also define the in-the-class CA branch which corresponds to the true label $y$ of given ROI and the remaining are the out-of-the class branches. The in-the-class and out-of-the-class CA branches are trained to achieve the cell classification through the selected positive and negative samples. Considering different classes are mutually exclusive, the detailed sample selection rule is given:

1. For the category of NILM, all the cell samples as the positive samples are fed into the NILM branch.
2. For the category of LSIL, $N$ positive and $N$ negative samples are fed into the LSIL branch where $N = n_c \times 10\%$. Besides, the $N$ positive samples are taken as the negative samples to input the NILM branch.
3. For the category of HSIL, $N$ positive and $N$ negative samples are fed into the HSIL branch where $N = n_c \times 10\%$. the $N$ positive samples are used as the negative samples to input the NILM and LSIL branches, respectively.

All the branches are trained through the selected samples and corresponding class-specific cross-entropy loss function. Therefore, the cell classification loss $L_{cell}$ is formulated as

$$L_{cell} = \frac{1}{3}(L_{NILM} + L_{LSIL} + L_{HSIL}) \tag{8}$$

where $L_{NILM}$, $L_{LSIL}$ and $L_{HSIL}$ are the class-specific cross-entropy loss function. The entire network is trained by the following loss function and $\lambda$ is the regularization parameter.

$$L = L_{ROI} + \lambda L_{cell} \tag{9}$$

## 3. Experiments

To evaluate the effectiveness of our method, we collect 900 ROIs of cervical cytology provided by Motic with 3 categories (NILM, LSIL and HSIL) shown in Fig. 2. Each category has 300 ROIs. The size of ROI ranges from $1024 \times 1024$ to $2048 \times 2048$. Note that the label of ROIs is only available. The cell images detected by YOLOv4 are resized to $224 \times 224$ and represented by the $d_f = 2048$ dimensional ResNet-50 features. The network is trained for 90 epochs by stochastic gradient descent (SGD) and the parameter $\lambda$ is set to 0.25. The initial learning rate is set to 0.0003 and divided by 10 for every 30 epochs. All the experiments are conducted on a computer with an Intel Core i7-7820X CPU of 3.60 GHz and 2 GPUs of NVIDIA GTX 2080Ti.

We compare the proposed method with the ROI classification methods based on RNN, GRU, LSTM, BLSTM, and BLSTM+ViT w/o $L_{cell}$, respectively. The ROI classification results are presented in Table 1. Note that the four methods (RNN, GRU, LSTM and Att-BLSTM) only use the global contextual features for ROI classification and BLSTM+ViT w/o $L_{cell}$ encodes the ViT features into the Att-BLSTM but without the cell classification loss $L_{cell}$. As shown in Table 1, Att-BLSTM is better than the conventional RNN, GRU and LSTM ROI classification methods since it uses the attention mechanism to capture the most

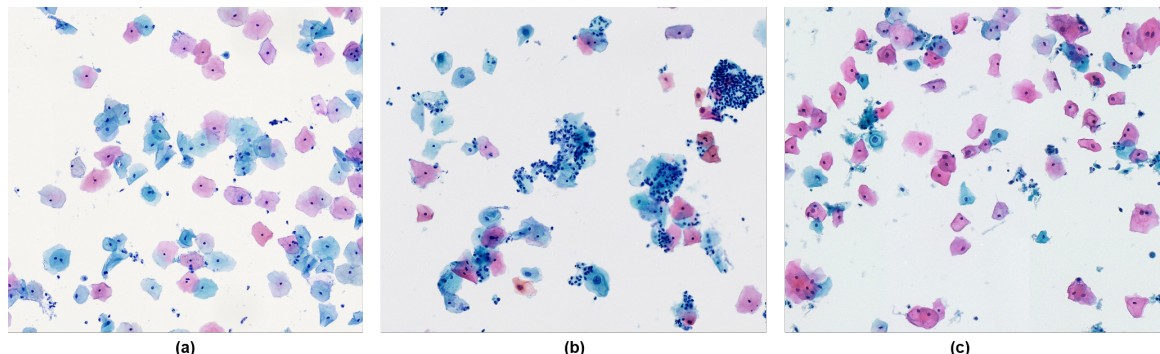

Figure 2: ROIs of cervical cytology for 3 categories: (a) NILM, (b) LSIL, (c) HSIL.

informative representation for the sequential data. As the ViT features of cells are encoded into the global Att-BLSTM features, the method BLSTM+ViT w/o $L_{cell}$ is superior to Att-BLSTM. It indicates the local ViT features contributes to improving the representational ability of global ROI features. More importantly, the proposed method outperforms other methods. It can be explained that our method achieves the trade-off between ROI and cell classification in a weakly supervised way and thus yields more discriminant ability.

Table 1: Comparison of ROI classification accuracies (%).

| Methods | RNN | GRU | LSTM | Att-BLSTM | Att-BLSTM + Vit w/o $L_{cell}$ | Our method |
|---------|-----|-----|------|-----------|-------------------------------|------------|
| Accuracy | 61.48 | 65.92 | 68.15 | 71.85 | 72.59 | 73.33 |

Moreover, the cells within 900 ROIs are annotated by the expert for the comparison of cell classification performance. 70% of cells from each category are used to train the ResNet-50 and the remaining cells are taken as the test set. As shown in Table 2, our method is comparable with the supervised ResNet-50. It indicates the weakly supervised learning of our method is beneficial to cell classification and generates promising classification performance without much manual cell-level annotation.

Table 2: Comparison of cell classification accuracies (%).

| Methods | RestNet-50(70% training) | Our method |
|---------|--------------------------|------------|
| Accuracy | 95.48 | 93.53 |

## 4. Conclusion

In this paper, we introduce a weakly supervised global-local learning for cervical cytology ROI and cell classification. It respectively applies attention-based bi-directional LSTM (Att-BLSTM) and the Vision Transformer (ViT) to explore the global contextual information of ROI and local cell patterns. The cross attention (CA) is used to encode local ViT cell features into the ROI representation. Consequently, the discriminative ability of ROI features is improved. More importantly, the CA score is taken as the pseudo label to weakly supervise

the cell classification through the in-the-class and out-of-the-class branches. Experimental results demonstrate our method has better ROI classification results and particularly shows the promising cell classification performance.

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
