# OpenReview forum: "Weakly Supervised Global-Local Feature Learning for Cervical Cytology Image Analysis"
_MICCAI.org/2021/Workshop/COMPAY — Reject_

### Official Review · Reviewer_Nmav · 2021-08-02
**looks promising, but needs more work**

**Rating:** 4
**Confidence:** 3

**Review:**

The authors propose a weakly supervised leraning approach combining LSTMs, Transformers and Attention to classify cervical cells from roi-level labels. The combination seems novel, but the details of the architectural blocks and their impact on performance are not clear. Specifially, in the first step "The cervical cells are firstly detected by YOLOv4". Does this mean that a pretrained yolo-model is used? How was this model trained? In the next step "We take the CNN features of cells as the sequential data and use the attention- based bi-directional LSTM (Att-BLSTM)...". So, a random(?) temporal order is imposed on the independent features? Are LSTMs really the best choice here? This seems to work, but I would like this to be explored in more detail. For the evaluation of the approach, I would suggest to use some kind of (sota) baseline and also to focus more on ablation studies to show that each block of the architecture has significant impact on the pefromance. Overall, more work and clear explanations are required so I can't recommend acceptance at this stage.

---

### Official Review · Reviewer_Xbn6 · 2021-08-11
**Relevant problem and solution but lacking novelty and comparative analyses**

**Rating:** 6
**Confidence:** 5

**Review:**

The paper introduces a weakly supervised global-local deep learning-based pipeline for cell classification in cervical cytology. It applies attention-based bi-directional LSTM, vision transformer, and cross-attention approaches to explore and encode global contextual information and local cell patterns. Experimental results on the Motic dataset illustrate the potential of the pipeline.

Pros:
- The paper addresses a relevant problem in cytology.
- The proposed weakly supervised approach is interesting.
- The experiments are performed on a published dataset.
- The paper is well written.

Cons:
- The discussion of previous works is shallow.
- The pipeline is a patchwork of existing pretrained models.
- The rationale behind the use of LSTMs is unclear.
- The experiments lack a comparison with the state of the art.

Specific comments:

- The title of the paper is too broad. Specifically, it claims to provide solutions to "cervical cytology
image analysis" in general, but later on it becomes clear that the focus is actually on cell classification. The title should be updated to reflect this focus. Also the abstract needs to be updated accordingly.

- In the introduction, the discussion of prior work in cervical image analysis, in particular cell classification, is very shallow. This needs to be improved. There is quite some body of literature on the subject: https://doi.org/10.1109/ACCESS.2020.2983186

- It is unclear a priori why the use of LSTMs would be beneficial for the purpose. As the CNN cell features are treated as the sequential data, it would seem that the results depend on the order of the features. This needs to be tested and discussed.

- In the description of the experiments, the Motic dataset should be properly cited (either a literature reference or a URL or both).

- More information is needed on whether and how much the cell images detected by YOLOv4 may vary in size. It is mentioned that the images are then all resized to 224 x 224 pixels. But if the initially detected images are not of the same size, it means the pipeline is throwing away some potentially very essential information, namely the size of the cells.

- It seems the proposed method is compared only with some of its own variants (ROI classification methods based on RNN, GRU, LSTM, BLSTM, and BLSTM+ViT w/o Lcell) but not with existing methods from literature or public challenges (summarized in the review paper linked above). The experiments need to be extended to provide a more comprehensive comparison with the state of the art.

---

### Decision · Program_Chairs · 2021-08-25

Reject